# The EBLM Project—From False Positives to Benchmark Stars and Circumbinary Exoplanets

**Pierre F. L. Maxted** [1,*] , **Amaury H. M. J. Triaud** [2] and **David V. Martin** [3]

1.  Astrophysics Group, Keele University, Stafforshire ST5 5BG, UK
2.  School of Physics and Astronomy, University of Birmingham, Edgbaston, Birmingham B15 2TT, UK; a.triaud@bham.ac.uk
3.  Department of Physics and Astronomy, Tufts University, 574 Boston Avenue, Medford, MA 02155, USA; david.martin@tufts.edu
*   Correspondence: p.maxted@keele.ac.uk

**Abstract:** The EBLM project aims to characterise very-low-mass stars that are companions to solar-type stars in eclipsing binaries. We describe the history and motivation for this project, the methodology we use to obtain the precise mass, radius, and effective temperature estimates for very-low-mass M dwarfs, and review the results of the EBLM study and those from related projects. We show that radius inflation in fully convective stars is a more subtle effect than what was previously thought based on less precise measurements, i.e., the mass–radius–effective temperature relations we observe for fully convective stars in single-line eclipsing binaries show reasonable agreement with the theoretical models, particularly if we account for the M-dwarf metallicity, as inferred from the analysis of the primary star spectrum.

**Keywords:** eclipsing binary stars; very-low-mass stars; fundamental properties of stars; exoplanets

## 1. Introduction

Very-low-mass stars, i.e., stars with a mass $M \lesssim 0.35 \, M_\odot$, are approximately the same size as the largest "hot-Jupiters" (HJs)—gas-giant exoplanets with masses and radii similar to that of Jupiter and orbital periods of a few days or less. This means that the transit of a solar-type star by either a hot Jupiter or a very-low-mass star will produce a dip in the light curve with a depth of about 1 % (e.g., Figure 1). Consequently, ground-based surveys that monitor the brightness of solar-type stars to discover transiting hot Jupiters also find many solar-type stars with very-low-mass stellar companions in eclipsing binary systems (EBs). The label "EBLM" (eclipsing binary–low mass) was used by members of the WASP project [1] ("Wide Angle Search for Planets") to flag these systems as "false positives". Despite their intrinsic faintness, field M dwarfs are attractive targets to identify rocky planets orbiting in the habitable zone of their host star, such as the seven planets orbiting the star TRAPPIST-1 ($M_\star \approx 0.08 \, M_\odot$) [2–4].

The EBLM project was first presented in an abstract submitted to the 2012 meeting of the American Astronomical Society by Hebb et al. [5] as "an ongoing program to examine the mass–radius relation of M dwarfs as a function of metallicity and activity using a large sample of EBs composed of an F, G or K dwarf primary star and an M dwarf secondary". The lack of good data for very-low-mass stars available at that time is demonstrated in Figure 2. The stars in this figure are the complete sample of M dwarfs in eclipsing binary systems available to a study published by Spada et al. in 2013 [6], comparing the mass–radius and mass–effective temperature relations for single low-mass stars (mass $\lesssim 0.7 \, M_\odot$) to low-mass stars in binary systems. That study also includes radius and effective temperature measurements for single M dwarfs from long-baseline interferometry in their discussion. Mass estimates for single M-dwarf stars are based on empirical relations, which complicates

the interpretation of these data, so we have decided not to include these measurements in this review.

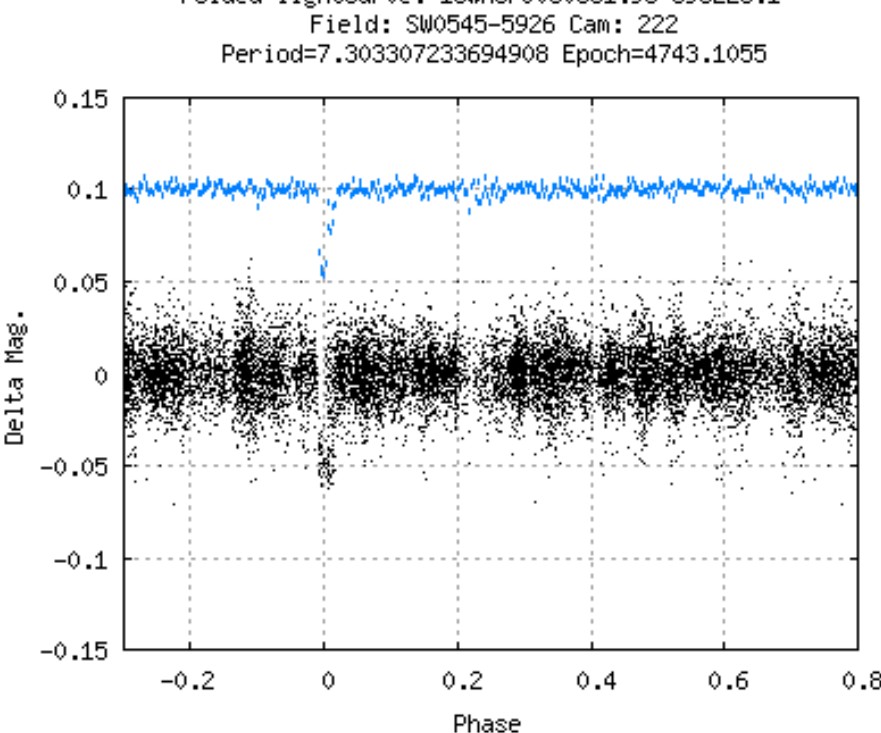

**Figure 1.** Phase-folded light curve of EBLM J0608-59 (TOI-1338, BEBOP-1) from the WASP project. The data in blue are phase-binned and offset vertically to make the transit at phase 0 easier to see in this automatically generated diagnostic plot. This system was first flagged as an EBLM system by Amaury Triaud on 2010-07-02 based on follow-up radial-velocity measurements obtained with the CORALIE spectrograph on the Swiss Euler 1.2 m telescope. This binary would later be discovered to host two circumbinary planets, one identified from a transit observed in the TESS light curve [7] and the other long-period planet identified from radial-velocity measurements obtained by the BEBOP project [8].

The study by Spada et al. [6] is just one of many studies of the "radius inflation problem", an issue identified by Hoxie as early as 1970 [9], whereby the radii of low-mass stars are observed to be larger than predicted by the models and the effective temperatures are underpredicted such that the predicted mass–luminosity relation is approximately correct. The same problem was noted by Popper [10], who described a discrepancy in the ages inferred for low-mass stars in eclipsing binary systems. The question of whether or not stars in eclipsing binary systems (EBs) are suitable for testing models of single stars has been widely debated because the data available for low-mass stars in EBs were dominated by studies of short-period systems ($P_{orb} \lesssim 3$ days). The advent of ground- and space-based photometric surveys made it much easier to find long-period EBs and their parameters appear consistent with short-period EBs [11]. The strong tidal interaction between the stars in a short-period binary forces them to rotate at or near to the orbital period, resulting in increased magnetic activity. This complicates the analysis of the light curves and the interpretation of the results obtained [12]. Magnetic activity has been implicated as the cause of the radius inflation problem [13–15], but the observational evidence for this hypothesis is ambiguous [11,16,17] and more than one physical mechanism has been suggested for this effect [13,18–20]. It is interesting to test theories of radius inflation against observations of EBLMs because stars with masses $\lesssim 0.35\,M_\odot$ lack a radiative core [21], so models that

successfully reproduce the properties of stars with masses $\sim 0.6\,M_\odot$ may not work for very-low-mass stars [22,23].

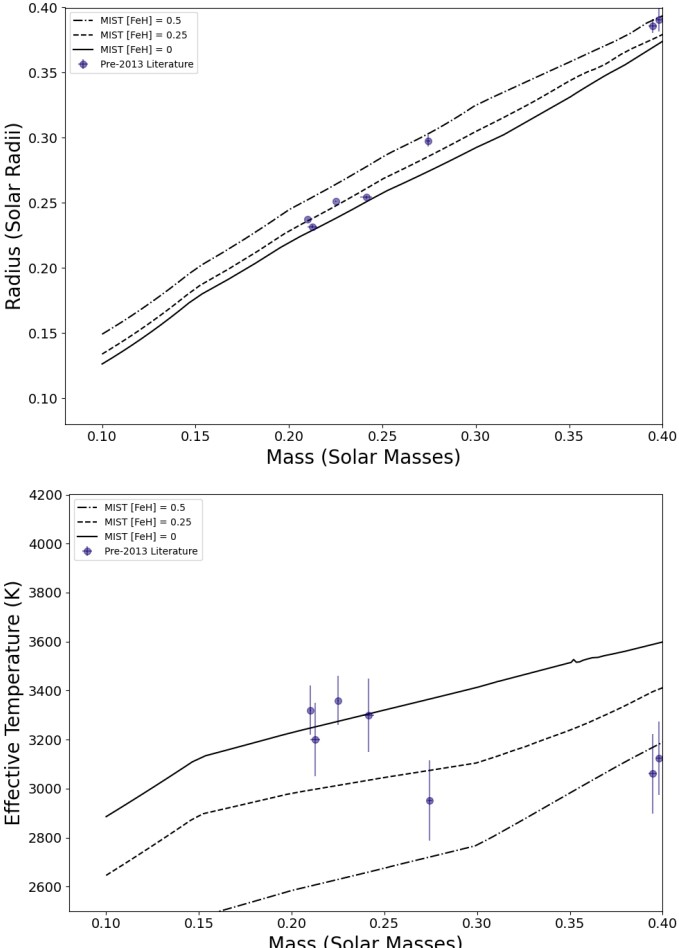

**Figure 2.** Mass, radius, and effective temperature measurements for very-low-mass stars in eclipsing binary systems published up to 2013. MIST isochrones are shown for metallicities as-labelled at an age of 10 Gyr [24,25].

The EBLM project has published 10 papers over the past decade in a series entitled "The EBLM project", in which we have used follow-up observations to characterise the stars in these eclipsing binary star systems [11,26–34]. The project is ongoing and will soon publish two more papers in this series [20,35]. In addition, members of the project have produced several studies of EBLM systems and similar systems identified in other surveys not published as part of the EBLM series [36–40].

In this review, we first describe the motivation and goals of the EBLM project, the techniques we have developed to characterise these binary systems, the highlights of the results we have obtained to date, and the implication of these results for our understanding of very-low-mass stars. We also look forward to the stellar and exoplanet science that can be achieved in the near future using the data from new surveys and instrumentation to observe these fascinating systems.

## 2. Goals of the EBLM Project

The main objective of the EBLM project is to produce an empirical mass–radius–luminosity–metallicity relationship for fully convective stars by measuring precise and accurate physical parameters for main-sequence M-dwarf stars with masses $\lesssim 0.35\,M_\odot$.

This goal has been achieved by observing M dwarfs that transit more massive solar-type stars in single-lined eclipsing binaries using high-precision photometry and high-precision radial-velocities. The mass and radius of the secondary low-mass companion can then be inferred given a mass or radius estimate for the primary star, either using stellar models calibrated on the Sun itself, or from a well-determined empirical relation. Very recently, we used GAIA radii for the primary and calculated a mass for the primary star from the eclipse model [20], to be as data-driven as possible. With high-quality photometry, the low-mass companion's luminosity can be also measured from the depth of the secondary eclipse. A key advantage of EBLM systems is that the metallicity of the binary system can be estimated from the spectrum of the solar-type star. As the project advanced, we developed new methods that make it possible to resolve some of the secondaries spectroscopically, i.e., to measure absolute masses and radii directly despite the large contrast ratio between the low-mass star and its solar-type primary [36].

There are two main scientific interests for which a mass–radius–luminosity–metallicity relation is needed for fully convective stars. The first is to improve the accuracy of mass and radius estimates for rocky terrestrial planets being discovered as transiting very-low-mass stars such as TRAPPIST-1 [3,41] and SPECULOOS-2 [42]. These systems provide the fastest means to explore the diversity atmospheres for rocky planets and likely represent the largest number of rocky planets in our Galaxy [4]. The second is to study the radius inflation problem for low-mass stars and to test whether it is related to magnetic activity and/or metallicity. The large contrast ratio prevents the M-dwarf companion from being observed directly in any detail directly, but the proximity of the binary can be used as a proxy for magnetic activity inferred from stellar rotation assuming tidal synchronisation, and the metallicity can be determined by analysing the spectrum of the solar-type primary star.

Currently, the EBLM project has collected high-precision radial-velocities on a sample of over 150 systems in the southern hemisphere and about 70 systems in the northern hemisphere. High-quality light curves are now available for many of these EBLM systems thanks to the TESS mission [43]. A subset of 100 of these binary stars with orbital periods >5 days are also being monitored intensively using radial-velocity measurements made with high-resolution spectrographs as part of the BEBOP survey (Binaries Escorted By Orbiting planets) for circumbinary planets [44,45].

The quality of the light curves produced via ground-based surveys like WASP are good enough to detect periodic dips in the light curves with depths $\gtrsim 0.5\,\%$, but lack the signal-to-noise ratio needed to characterise the exact shape of these eclipses (see Figure 1). This makes it difficult to distinguish EBLM systems with total eclipses from eclipsing binaries with shallow partial eclipses with these data. Partial eclipses contain less information than total eclipses, and so the analysis of these light curves produces results that are less reliable and less precise than those obtained for EBLM systems [32]. We also find that the companions to the primary stars in these partially eclipsing binary systems are typically stars with masses $>0.35\,M_\odot$, so we use follow-up observations to avoid these binaries, if possible.

Secondary but important objectives of the EBLM projects involve a range of additional scientific topics:

- A comparison sample to the hot Jupiter population. Hot Jupiters were the first exoplanets detected with the radial-velocity and transit methods [46,47]. Despite their relatively low occurrence rates (circa 1 per 200 solar-type stars [48,49]), they are the easiest to find and the most studied [50]. Many of their early characteristics were unclear and some of their characteristics are still puzzling today [50]. Most hot Jupiters are, for instance, inflated beyond what irradiative models predict [51]. When we initiated the EBLM project, some of the hot Jupiters were found on small but non-zero eccentricities, leading to questions about tidal circularisation [52–55].[1] The spin–orbit alignment for many transiting hot Jupiters have been studied using measurements of the Rossiter–McLaughlin effect. A large fraction of these planets are found to have orbital planes misaligned with their host star's equatorial plane.

A likely explanation came in with the Kozai–Lidov mechanism [57–59], which is also invoked for the production of short-period binaries [58]. EBLM systems have radii and effective temperatures similar to hot Jupiters, i.e., they occupy a similar region in a colour-magnitude diagram to hot Jupiters [60,61]. As such, EBLM systems provide an interesting comparison sample for many properties of the hot Jupiter population. Using this comparison sample, our early hope was to ascertain whether trends discovered about hot Jupiters might be caused by observational biases, or instead explained via physical processes common to all types of objects, e.g., to explore the apparent radius inflation reported in some early studies of hot Jupiters, brown dwarfs, and low-mass stars.

- Refining the boundaries of the brown dwarf desert. Brown dwarfs are star-like objects with masses between 13 and 80 $M_{Jup}$ that fuse deuterium in their core but are not massive enough to fuse hydrogen [62]. They are rarely found in eclipsing geometries, so their physical parameters were mostly unknown when the EBLM project started in 2008. Radial-velocity surveys had shown the existence of a "brown dwarf desert"—a reduced occurrence rate of brown dwarf companions to solar-type stars [63]. Although the occurrence rate is low, it is not zero, so it was expected that some brown dwarfs would eventually be identified in eclipsing binary systems. Transit surveys can identify hot Jupiters in larger numbers than radial-velocity surveys, so we expected that the WASP survey would be an effective way to find these eclipsing brown dwarf systems. The EBLM project could then improve the study of the brown dwarf desert by providing a greater resolution on its shape, of which its masses are the least frequent, and whether the bounds of the desert depend on the mass of the primary star. Since the EBLM systems are typically at a short period, this also refines the characteristics of the desert at a short orbital period. Preliminary results on this topic appeared in [29].

- The study of tides. The phenomenon of tides has been known since antiquity and has many visible effects in the Solar System, e.g., Mercury's spin–orbit resonance, tidally induced volcanic activity on Jupiter's moons, and of course, the effects of our own Moon. The loss of energy from the orbit due to tides leads towards the lowest energy configuration in which the orbit is circular and the rotation of the stars is aligned and synchronised with the orbit [53]. There is limited observational evidence available to study the efficiency of orbital synchronisation and circularisation and a function of orbital separation and companion mass. The canonical orbital period for circularisation is ≈8 days [64,65]. This result is based on the samples dominated by binary systems with stars of similar masses. Furthermore, more precise radial velocities are needed to probe if an orbit is truly circular, or instead, if a small but non-negligible eccentricity remains [29]. Tidal synchronisation can be probed using light curves to measure the rotation period of the primary star either via starspot modulation [66] or using spectroscopy to infer the rotation period via spectral line broadening [67]. More work is needed that combines both methods to resolve the ambiguities that exist from using only one of these methods. The alignment of the spin and orbital axes can be measured using the Rossiter–McLaughlin effect. Many results using this method have been published for hot Jupiters and, perhaps surprisingly, often find the rotation and orbital axes to be significantly misaligned [59,68]. Similar studies of binary stars have largely been restricted to massive stars ($>2M_\odot$, [69]). These results may not be representative of the tidal efficiency in less massive stars which have deep convective envelopes, in contrast to the radiative atmospheres of more massive stars. For all of these tidal signatures, EBLM can more easily be used to probe the limit of efficient tidal interactions, since the low-mass secondaries have a smaller influence at a given separation.

- A sample to seek circumbinary planets. The observation of circumbinary planets was first proposed by [70–72], before the discovery of 51 Peg b in 1995 kick-started the exoplanet revolution [46]. Despite theoretical work implying such planets might not exist or may be very rare, there were ample reasons to test these predictions, given

that exoplanet detections have often defied theoretical expectations, e.g., the existence of both hot Jupiters and super-Earths were unexpected discoveries. The first proposed method for circumbinary detection was the transit method [70–72], which led to the first bona fide circumbinary planet discovery (Kepler-16 [73]). Since the early 2000s, there have also been significant efforts to use radial velocities to find circumbinary planets, e.g., the TATOOINE project (The Attempt To Observe Outer-planets In Non-single-stellar Environments). This project's goal was to detect circumbinary planets orbiting bright nearby double-lined binaries [74,75]. However, despite much effort spent in refining algorithms to both disentangle the spectra and extract accurate radial-velocities, the TATOOINE team concluded that too much noise remained present in their data for an effective circumbinary exoplanet search [75]. Instead, they suggested that single-lined systems would be better suited for such surveys [75]. Few suitable binary systems were known at the time, but the advent of large-scale surveys for transiting exoplanets, such as WASP that identified dozens of EBLM systems, made this idea viable. In particular, the BEBOP search for circumbinary planets (Binaries Escorted By Orbiting Planets) is an offshoot of the EBLM project [44], which we describe in more detail in Section 5.1.

### 3. History and Background to the EBLM Project

#### 3.1. The WASP Project

The WASP survey used two robotic telescope mounts, each holding an array of eight cameras to search for transiting exoplanets orbiting stars in the magnitude range $V \approx 9 - 12$ [1].[2] One instrument was sited at the Observatorio del Roque de los Muchachos, La Palma, and the other at Sutherland Observatory, South Africa. The cameras were thinned e2v charge-coupled device (CCD) detectors with $2048 \times 2048$ pixels mounted on 200 mm f/1.8 lenses with a broad-band (400–700 nm) filter. From July 2012, the instrument in South Africa was equipped with 85 mm, f/1.2 lenses and r′ filters to target brighter stars than could be observed with the 200 mm lenses [76]. Observations of various target fields were obtained every 5–10 min using two 30 s exposures. The data obtained were automatically processed and analysed in order to identify stars with light curves that contain transit-like features that may indicate the presence of a planetary companion [77]. Light curves such as the example shown in Figure 1 were generated using synthetic aperture photometry with an aperture radius of 48 arcsec (112 arcsec for the 85 mm lenses). Following a pilot survey with four cameras at the northern instrument in 2004, both instruments operated almost continuously from 2006 to 2014. These observations have led to the discovery of almost 200 transiting exoplanets [78] and have been used to discover and characterise a wide variety of variable stars, including stripped red giant stars [79,80], an extrasolar ring system transiting a young solar-type star [81], and hundreds of pulsating Am stars [82].

The WASP survey is one among several ground-based transit surveys that have started post-2000 that have successfully confirmed transiting exoplanets [83–91]. All of these surveys produce many "false positives" [92–95] and other teams have also produced some efforts to characterise "EBLM-like" systems from these surveys, often with similar aims to the EBLM project itself [96–101].

#### 3.2. The WASP Follow-Up Programme and the Origin of the EBLM Flag

Identifying a feature in the light curve consistent with the properties of a transiting planet is not enough to claim a confirmed planet detection. For ground-based instruments like WASP, the large photometric aperture means that there are a large number of false astrophysical positives that have to be weeded out, in addition to false alarms created by imperfect data or astrophysical noise (intrinsic stellar variability, blended eclipsing binary stars, etc.). WASP used a variety of flags to identify which of the candidates are false alarms, false positives, or confirmed planets. Like most transit surveys (including space-based surveys like CoRoT, Kepler, and TESS), the WASP project relies on a follow-up programme involving ground-based telescopes to distinguish between these possibilities [102]. WASP's

standard procedure was to first create a list of 10 to 30 transiting planet candidates that were passed on to the follow-up team, with new lists created a few times per year. Such lists were created by selecting candidates from a long catalogue of stars found automatically to show periodic transit-like features in their light curves [77]. The WASP project invested a lot of effort to create a web-based portal where all the information required to select and organise the follow-up were included. Information about each candidate was available on a web page automatically generated from the WASP database. These web pages would show a plot of phase-folded WASP photometry, revealing the transit-like signal (e.g., Figure 1), a periodogram of the same data, and additional information such as stellar colour, images of the field of view, etc. [102]. An extremely helpful feature of this system was a comment section where the vetting team and follow-up observers could communicate about a particular target, share data, and assign a number of flags that would determine the next course of action. Any post in the comment section would trigger an email alert to the relevant members of the follow-up team, enabling them to make the best use of the available observing time, accounting for the latest information available on their targets.

For candidates with other stars of similar brightness nearby, one or more photometric measurements in and out of transit ("on-off" observations) were used to check whether the signal was indeed due to a transit-like feature in the light curve. In many cases, background eclipsing binaries were responsible for the transit-like signal and were flagged as "BEB", or the shape of the transit in the higher-quality light curve showed that the target was itself an eclipsing binary ("EB"). For candidates that passed these tests, or if the WASP photometry showed a well-defined, box-like event, on a well-isolated star, radial velocities were collected at the expected extremes of the Doppler reflex motion using high-resolution spectrographs. The main instruments used for follow up were CORALIE in the southern hemisphere and FIES and SOPHIE in for the northern hemisphere. Stars identified as double-lined (SB2) binaries were removed from the follow-up programme and flagged as "EB". Stars showing a single set of spectral lines with radial-velocity variations of a few $km\,s^{-1}$ (SB1), implying a companion mass too large for an exoplanet, were classified as an "EBLM"—Eclipsing Binary, with a Low-Mass companion [102].

### 3.3. Origins of the EBLM Project

A few events triggered the start of the EBLM project, i.e., a systematic exploitation of the WASP data to identify solar-type stars with very-low-mass stellar companions. First, WASP identified a massive short-period planet, WASP-18 b [103], and there was a concern that others like it might have been missed. Secondly, a few eclipsing brown dwarfs were identified via other surveys from 2008 onwards [104,105]). Thirdly, we wanted to collect a comparison sample to understand the growing evidence from observations of the Rossiter–McLaughlin effect that showed a large fraction of hot Jupiters occupying inclined orbits, with respect to the stellar rotation axis [59] (and see Section 2).

So, from mid-2008, the radial-velocity observations were intensified on candidates flagged as EBLM to make sure brown dwarfs and planets like WASP-18 b were not being missed. This led to the discovery of the transiting brown dwarf WASP-30 b [106].[3] Meanwhile, the EBLM project started to take shape as its own sub-programme within the wider WASP consortium. The main selection criteria for this project were that the star should be confirmed as an eclipsing or transiting system and the semi-amplitude of the Doppler reflex motion should be <100 $km\,s^{-1}$. With this setup, transiting exoplanets can be identified, brown dwarfs can be detected, the paucity of the brown dwarf desert could eventually be established, fully convective stars' physical properties could be studied and compared to stellar evolution models, and the orbital elements between transiting planets ($e$ the eccentricity and $\lambda$ the spin–orbit angle[4]) could be compared to low-mass eclipsing binaries in order to study the influence of tides. The EBLMs were observed at the same time and in the same manner as the transiting exoplanets, with the purpose of creating as homogeneous a sample as possible. The only distinction between the EBLM sample and the exoplanet and brown dwarf samples was in the number and quality of the radial-velocities.

While planets typically received as many 1800 s exposures as was needed to confirm them, the EBLM targets typically were observed with 600 s exposures. We decided to obtain a minimum of 13 measurements over two or more observing seasons to make sure the orbital phases were covered multiple times in order to produce robust orbital and physical parameters. The first paper of the EBLM project describes many of these themes and early hopes [26].

## 4. Methodology

### 4.1. Mass and Radius Estimates for EBLM Systems

The first defining characteristic of an EBLM system is a drop in the flux of about 1% that lasts for a few hours and repeats with a period of a few days or more, with no other features in the optical light curve detectable using photometry of the quality provided by the WASP project. The shape, depth, and width of the dip in the light curve contains information on the geometry of the binary system. There are many software packages that are available that can be used to extract this information by fitting a model to the light curve, e.g., pycheops [107], jktebop [108], allesfitter/ellc [109,110], exofast [111,112], batman [113], TLCM [114], etc. Figure 3 shows the geometry of the transit for an EBLM system with an impact parameter $b \approx 0.36$. If the orbit is circular (eccentricity $e = 0$), then the impact parameter is related to the orbital inclination, $i$, by the equation

$$b = \frac{a}{R_1} \cos i,\tag{1}$$

where $a$ is the semi-major axis of the binary orbit and $R_1$ and $R_2$ are the radii of the primary star and the low-mass companion, respectively. Synthetic light curves for a hypothetical EBLM system with a solar-type primary star and an orbital period of $P = 3.3$ days are shown in Figure 4 for a range of impact parameter values. The depth of the transit is determined mainly by the fraction of the primary star's apparent disk that is covered by the low-mass companion, i.e., depth $\approx k^2 = (R_2/R_1)^2$. If the orbit is circular then the time from the first to fourth contact, it is

$$t_4 - t_1 \approx \frac{P}{\pi} \frac{R_1}{a} \sqrt{(1+k)^2 - b^2}.\tag{2}$$

The variation in the specific intensity emitted by the primary star across the stellar disk is known as a centre-to-limb variation (CLV) or limb darkening. This is typically assumed to be a simple function of the cosine of the angle between the line of sight and the normal to the surface, $\mu = \sqrt{1 - r^2}$, where $r$ is the distance from the centre of the stellar disk to the limb, relative to the stellar radius, i.e., $r = 1$ at the limb. Limb darkening is seen as a curvature in the light curve between the second and third contact points. In general, limb darkening is a stronger effect at shorter wavelengths. At a given wavelength, the main parameter that determines the strength of limb darkening is the stellar effective temperature, with the effect being stronger for cooler stars. The CLV at wavelength $\lambda$, $I_\lambda(\mu)$, is typically parameterised using the quadratic limb-darkening law—

$$I_\lambda(\mu)/I_\lambda(1) = 1 - u(1 - \mu) - v(1 - \mu)^2\tag{3}$$

or Claret's 4-parameter law [115]—

$$I_\lambda(\mu)/I_\lambda(1) = 1 - c_1(1 - \mu^{\frac{1}{2}}) - c_2(1 - \mu) - c_3(1 - \mu^{\frac{3}{2}}) - c_4(1 - \mu^2);\tag{4}$$

or the power-2 law—

$$I_\lambda(\mu)/I_\lambda(1) = 1 - c(1 - \mu^\alpha).\tag{5}$$

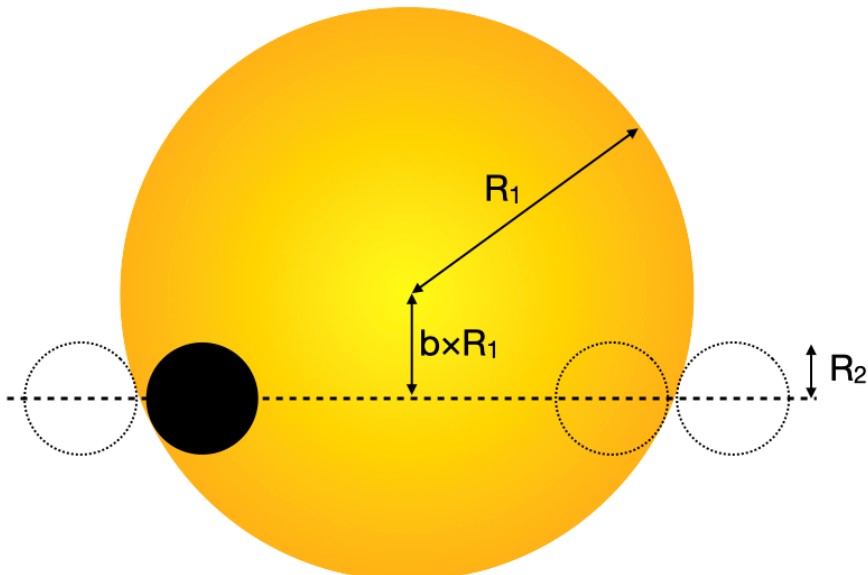

**Figure 3.** Geometry of a transit for a solar-type star (yellow disk) with a low-mass companion (black disk) at the second contact point. The dotted lines show the position of the low-mass companion at the first, third, and fourth contact points (from left to right). The dotted line show the transit path of the low-mass companion.

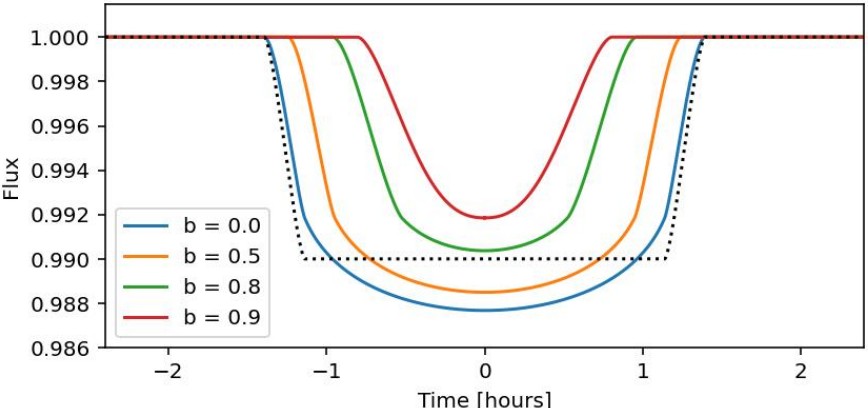

**Figure 4.** Model transit light curves for EBLM systems with $k = 0.125$, $R_1/a = 0.1$, and $P = 3.3$ days at optical wavelengths for various values of $b$, as noted in the legend. The dotted line shows a transit for the same system with $b = 0$ and no limb darkening.

The power-2 law performs better than the quadratic law in terms of more accurately recovering the correct value of $k$ in simulated light curves [116]. In general, the limb darkening data computed from stellar model atmospheres agrees well with the measurements made on transiting exoplanet host stars [117]. Small differences between the observed and computed CLV are seen that can be ascribed to the effect of the magnetic field that is expected in the photospheres of these solar-type stars [118].

There are two simplifications that have been made in the description above. Firstly, we have neglected the flux from the companion star. This typically has a very small effect on the transit depth because the flux ratio at optical wavelengths is $\ell \sim 0.1\%$. The occultation of the companion produces a secondary eclipse in the light curve with a depth $1/(1 + 1/\ell) \approx \ell$. The depth of this eclipse can be measured with high-quality photometry and can be used to estimate the effective temperature of the companion star (Section 4.3) [37]. The width and orbital phase of the secondary eclipse are related to the orbital eccentricity and the longitude of periastron for the primary star's orbit, $\omega$. For small values of $e$, the phase of

mid-secondary eclipse is approximately $0.5 + e \cos \omega$. Limb darkening has no impact on the depth of a total eclipse and has a negligible effect on the short ingress and egress phases of the secondary eclipse in EBLM systems, so it is common to ignore limb darkening on the M dwarf for the analysis of these light curves.

Secondly, we have assumed that both stars are spherical. In some short-period EBLM systems, the distortion of the primary star by its companion can be seen as the "ellipsoidal effect"—a smooth variation in a flux with two maxima per orbital cycle between the eclipses. The semi-amplitude of this effect can be estimated using the approximation

$$A_{\text{ellip}} \approx \alpha_{\text{ellip}}\, q \left( \frac{R_1}{a} \right)^3, \tag{6}$$

where $q = M_2 / M_1$ is the mass ratio and $\alpha_{\text{ellip}} \approx 2$ for solar-type stars [119]. Other effects described in Ref. [119] (reflection effect and Doppler beaming) are negligible for EBLM systems. The value of $A_{\text{ellip}}$ as a function of the orbital period assuming $M_1 = 1\, M_\odot$ for various values of $R_1$ and $q$ is shown in Figure 5. An ellipsoidal effect with a peak-to-peak amplitude $\approx 0.5\%$ is detectable in WASP light curves, so binaries with orbital periods shorter than $P \approx 2$ days are clearly not candidate hot-Jupiter systems and so were typically flagged as eclipsing binaries or some other variable type, but not as EBLM systems. The low-mass companion will also be tidally distorted by the gravitational field of the primary star, but this has a negligible effect on the light curve because the oblateness of the companion is $\sim 5$ ppm for $P = 2$ days [120].

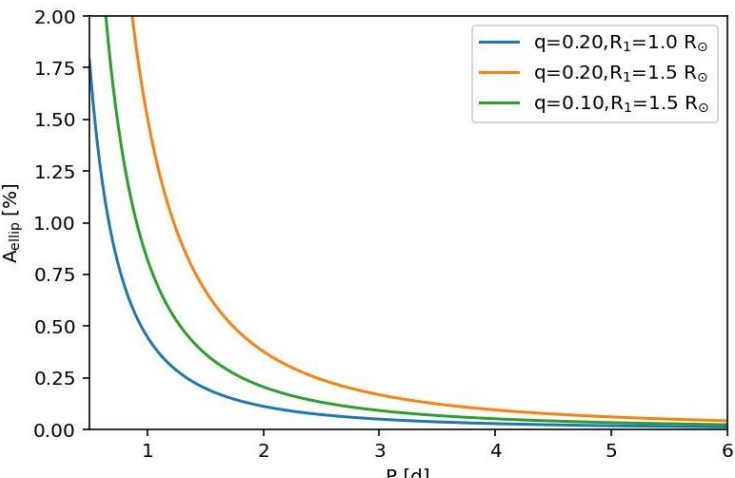

**Figure 5.** Semi-amplitude of the ellipsoidal effect from EBLM systems as a function of orbital period.

The second defining characteristic of an EBLM system is that the radial velocity of the primary star varies by $\gtrsim 10$ km/s during the orbital cycle. This shows that the companion is too massive to be an exoplanet. For a two-body Keplerian orbit, the radial velocity of the primary star as a function of the orbital phase is given by

$$V_r = K_1 \left[ \cos(\nu + \omega) + e\, \cos(\omega) \right], \tag{7}$$

where $\nu$ is the true anomaly obtained by solving Kepler's equation [121] and

$$K_1 = \frac{2\pi a \sin i}{(1+q)P\sqrt{1-e^2}}. \tag{8}$$

The radial velocity (RV) can be measured to a precision $\sim 10$ m/s using échelle spectrographs on small- to medium-sized telescopes [102]. Two RV measurements near orbital phases 0.25 and 0.75 are sufficient to estimate $K_1$ by assuming a circular orbit if the period and time of mid-transit have been measured from the light curve. Estimates of $e$ and $\omega$ can

also be made using a least squares of a Keplerian orbit if four or more RV measurements are available. The values of $K_1$ and $P$ are related to the stellar masses by the mass function—

$$f_m = \frac{M_2^3 \sin^3 i}{(M_1 + M_2)^2} = \frac{1}{2\pi G} K_1^3 P \left(1 - e^2\right)^{3/2}. \tag{9}$$

An estimate of the primary star mass, $M_1$, and a value of $\sin i \approx 1$ from the analysis of the light curve can then be used to estimate the companion star mass, $M_2$.

A very useful fact that helps when we are trying to make an estimate of the primary star mass is that the mean density of the primary star can be estimated directly from the following equation—

$$\rho_\star = \frac{3M_1}{4\pi R_1^3} = \frac{3\pi}{GP^2(1+q)} \left(\frac{a}{R_1}\right)^3. \tag{10}$$

This does require an estimate of the mass ratio, $q = M_2/M_1$, but a guess for $M_1$ that is wrong by 10% will lead to an error $< 1\%$ in $\rho_1$ for a typical EBLM system if $q$ is estimated via the mass function because the mass ratio is weakly dependent on the assumed primary star mass if $M_1 \gg M_2$ and the mass ratio only enters the equation for the stellar density via the term $(1+q)$. The primary star mass can then be estimated using an empirical relation [122] or by comparison to isochrones from a grid of stellar models [40]. In either case, an estimate of the primary star's effective temperature, $T_{\text{eff},1}$, and metallicity, [Fe/H], from an analysis of the stellar spectrum is required.

Another useful quantity that can be determined independently of any estimate of the primary star mass is the surface gravity of the companion [123]—

$$g_2 = \left(\frac{2\pi}{P}\right)^{4/3} \frac{(G f_m)^{1/3}}{(R_2/a)^2 \sin i}. \tag{11}$$

*4.2. Primary Star Characterisation*

In general, an accurate estimate of the primary star's mass and radius is needed in order to estimate the mass and radius of the M-dwarf companion. If a high-quality light curve is available, then the constraint on the mean stellar density of the primary star from Equation (10) can be used to estimate the primary star radius if a good mass estimate is available, or vice versa [20]. An accurate estimate of the primary star's metallicity, [Fe/H], will help to improve the accuracy of these mass and radius estimates and can be used to investigate the impact of metallicity on the mass–radius relation for M dwarfs if we assume that the [Fe/H] measurement for the primary star is indicative of the initial composition for both stars in the system.

In the early days of the project, there was little information available on the properties of the primary stars in the EBLM sample. Where follow-up observations to obtain high-quality spectroscopy were available, standard spectroscopic analysis techniques could be employed to estimate $T_{\text{eff},1}$, $\log g_1$ and [Fe/H] for the primary star [26–28]. Photometric techniques were employed to estimate $T_{\text{eff},1}$ for the sample of over 100 EBLM systems studied in Paper IV [29]. With no information on the value of $\log g_1$ or the parallax for these stars, masses had to be estimated assuming that these are main-sequence stars. The spectra used to measure the radial velocities of the stars in this sample typically have a low signal-to-noise ratio. Nevertheless, useful estimates for $T_{\text{eff},1}$ and [Fe/H] could be extracted from these data using a method based on wavelet analysis [39]. This method gives only loose constrains on $\log g_1$, but this was not a problem for the analysis of systems for which good-quality light curves were available because the stellar density estimated using Equation (10) could be combined with the estimates of $T_{\text{eff},1}$ and [Fe/H] from the analysis of the spectra to obtain a precise estimate of the primary star mass using stellar models [11,30].

There are a variety of analysis techniques and stellar models that can be used to infer stellar properties from high-resolution spectra. There can be significant differences in

the results obtained from different methods and models for the same star, particularly for metallicity estimates [124]. This is partly due to systematic errors in $T_{\text{eff},1}$ estimates for FGK stars, which is a problem for many areas of stellar and Galactic astrophysics, so extensive efforts have been made to establish a set of "benchmark stars" that can be used to calibrate methods to measure this quantity [125]. Methods to measure $T_{\text{eff},1}$ are based on the definition of effective temperature in terms of a star's luminosity—$L = 4\pi R^2 \sigma_{\text{SB}} T_{\text{eff}}^4$, where $\sigma_{\text{SB}}$ is the Stefan–Boltzmann constant and $R$ is the stellar radius [126]. If we divide both sides of this equation by $d^2$, where $d$ is the distance to the star, we can obtain the following equation—

$$T_{\text{eff}}^4 = \frac{4\mathcal{F}_{\oplus,0}}{\theta^2 \sigma_{\text{SB}}}, \tag{12}$$

where $\theta = R/d$ is the stellar angular diameter and $\mathcal{F}_{\oplus,0}$ is the bolometric flux, from the star at the top of the Earth's atmosphere, corrected for interstellar extinction. Until recently, it has only been possible to measure $\theta$ for dwarf FGK stars using long-baseline interferometry for a few bright, nearby stars. Repeated measurements of $\theta$ typically differ by about 50 μas, i.e., 5% for the majority of existing benchmark stars with $\theta \approx 1$ mas [127]. The advent of accurate parallaxes for millions of stars from the GAIA mission [128] has made possible a new method to accurately measure $T_{\text{eff}}$ for stars in eclipsing binary systems [129]. This method uses flux ratios measured at several wavelengths plus empirical $T_{\text{eff}}$–colour relations to determine the bolometric fluxes for both stars in the binary from their combined bolometric flux. Systematic errors in $R$ and $d$ are much less of an issue for stars in nearby eclipsing binary systems that have been analysed using high-quality data [130,131]. For EBLM systems, the bolometric flux from the M dwarf is almost negligible, and so this method can be applied to estimate $T_{\text{eff}}$ for the primary star given a rough estimate of the M-dwarf's properties if an accurate estimate for the primary star's radius is available. It is now feasible to measure the orbital velocity of the M dwarf in EBLM systems using high-resolution near-infrared spectroscopy. This gives a model-independent measurement of the primary star mass from which the primary star's radius can be derived via the mean stellar density using Equation (10) [20]. This method was first demonstrated using observations with the SPIRou spectrograph on the Canada–France–Hawaii telescope for the eclipsing binary EBLM J0113+31, together with the light curves from TESS and CHEOPS to measure $T_{\text{eff},1} = 6124$ K $\pm$ 50 K for the primary star in this binary [36]. EBLM systems with direct $T_{\text{eff},1}$ measurements obtained using this method are almost ideal as benchmark stars because they are within the magnitude range that can be directly observed via instruments such as 4MOST, WEAVE, etc. [132,133], i.e., they can be used for "end-to-end" tests of the accuracy of $T_{\text{eff}}$ and $\log g$ measurements published by these large-scale spectroscopic surveys.

*4.3. M-Dwarf Effective Temperature Estimates*

With high-precision photometry, it is possible to measure the depth of the secondary eclipse in the light curve of an EBLM system [32]. This depth gives a direct measurement of the flux ratio, $\ell_{\text{lambda}} = F_{\lambda,2}/F_{\lambda,1}$, where $F_{\lambda,2}$ is the flux from star 2 at the effective wavelength $\lambda$ of the instrument, and similarly, for $F_{\lambda,1}$. If we also have the ratio of the stellar radii, $k = R_2/R_1$, from the analysis of the transit, we can calculate the surface brightness ratio $J = \ell/k^2$.

The surface brightness of a star at some effective wavelength $\lambda$ is a smooth function of $T_{\text{eff}}$ that can be computed from stellar model atmospheres, with little dependence on other parameters. This means that the measurement of $J$ from the secondary eclipse and transit depths in the light curve gives a strong constraint on the effective temperature ratio between the two stars, i.e., $T_{\text{eff},2}$ can be estimated given an estimate of the primary star's effective temperature, $T_{\text{eff},1}$. Measurements of $T_{\text{eff},2}$ are valuable as both an addition test of models for low-mass stars and as a way to investigate the claim that radius inflation for low-mass stars also produces a lower effective temperature than expected in such a way that the mass–luminosity relation predicted by stellar models is approximately correct [9,134].

An early attempt published in Paper II to measure $T_{\text{eff},2}$ using J-band photometry from a ground-based telescope found an effective temperature $\sim 600\,\text{K}$ hotter than predicted by theoretical models for the 0.19-$M_\odot$ companion to EBLM J0113+31 [27]. This result was not supported by an analysis of the data from the TESS satellite, which found a value for $T_{\text{eff},2} \approx 3200\,\text{K}$, consistent with the expectations of the theoretical models [37]. This result demonstrates the difficulty in measuring these very shallow eclipse depths from the ground. This study also investigated the impact of using different models to calculate a surface brightness–effective temperature relation for the M dwarf and found that this introduces a systematic error $\sim 50$ K (see their Figure 3). Anomalous results were also found of the M-dwarf companions to KIC 1571511 and HD 24465 using the data from the Kepler telescope [135,136]. A more recent analysis of the same data that addressed some of the issues with the methods used in the previous studies does not support these anomalous results, but again, it found results consistent with the mass–$T_{\text{eff}}$ relation for other M dwarfs and a good agreement between the values of $T_{\text{eff},2}$ derived using the Kepler and TESS data [38].

Given the apparent difficulty in making accurate measurements of $T_{\text{eff},2}$, we designed an observing programme to measure eclipse depths for a sample of 23 EBLM systems as part of the CHEOPS Guaranteed Time Observation programme [137]. Where possible, we also analysed the data from the TESS mission for these targets in order to check for consistency in the $T_{\text{eff},2}$ measurements from the two instruments. First, the results from this project were presented in Paper VIII [32]. The results for a further five systems were presented in Paper IX [33]. This project also provides precise mass and radius estimates for the M-dwarf companions to the selected EBLM systems and benefits from a uniform methodology to determine the properties of the primary stars, e.g., a consistent metallicity based on an analysis of the available spectra by the CHEOPS "TS3—Target Characterisation" working group. The results using the CHEOPS and TESS light curves in these studies show good consistency with one another and tend to find $T_{\text{eff},2}$ as slightly higher than observed in other M dwarfs. The final analysis of the complete sample will include a consideration of the systematic errors due to moderate levels of magnetic activity observed on some of the primary stars [35].

### 4.4. The Rossiter–McLaughlin Effect

The formation mechanism for short-period planets and binary star systems may leave an imprint on the distribution of obliquities for their orbits. This can be studied using measurements of spin–orbit misalignment via the Rossiter–McLaughlin effect [59,138]. The absorption lines in the spectrum of the primary star are broadened by its projected rotation velocity, $v_{\text{rot}} \sin i_\star$. Note that the inclination of the star's rotation axis, $i_\star$ is not necessarily the same as the orbital inclination, $i$. The (mis-)alignment of the orbital and stellar rotation axes can be studied using the Rossiter–McLaughlin (RM) effect. This is a deviation from a Keplerian orbit observed in the RV measurements of the primary during the transit. It is caused by the asymmetry in the line profiles introduced by the companion blocking parts parts of the stellar surface that vary in radial velocity across the apparent stellar disk. The amplitude of the RM effect is approximately [139–141]

$$\Delta V = v_{\text{rot}} \sin i_\star k / (1 + k), \tag{13}$$

The calculation of the RM effect is not straightforward because it depends on details such as the instrumental resolution, the intrinsic line profiles, etc. [142]. For this reason, the preferred method to study the RM effect is to analyse the line profiles directly rather than the RVs [31]. The first EBLM target to receive an RM measurement was EBLM J1219-39 [26]. A more recent and higher-precision example was observed in the system EBLM J0608-59 [31]. In Figure 6, we show its data in comparison with the models computed with `ellc` [109].[5] We have obtained the necessary data to measure the RM effect for more than 20 EBLM systems and an analysis of these systems is ongoing.

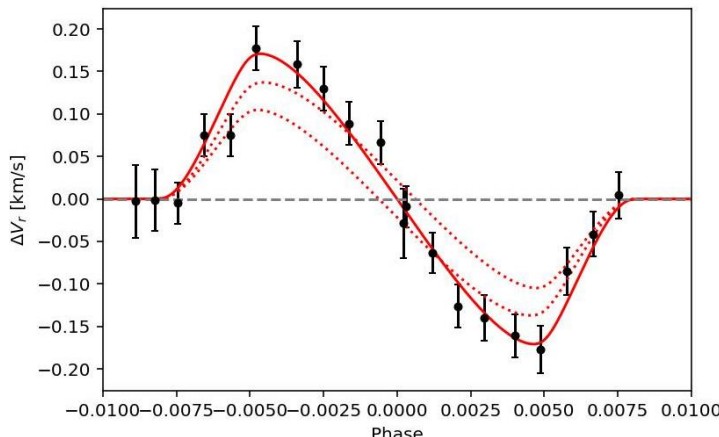

**Figure 6.** Radial-velocity (RV) measurements for EBLM J0608-59 (=TOI-1338, =BEBOP-1) showing the Rossiter–McLaughin (RM) effect in a system with a projected spin–orbit misalignment angle $\beta = 3 \pm 17°$ [31]. Red lines show the RM effect computed for $\beta = 0 \pm 45°$. Orbital phase is relative to the time of mid-transit. The best-fit Keplerian orbit has been subtracted from the RV measurements so that the RM effect can be seen more clearly.

## 5. Related Projects

### 5.1. The BEBOP Radial-Velocity Search for Circumbinary Planets

There were early claims of circumbinary planets around stellar remnants and post-common envelope binaries [143,144], but the discovery of circumbinary planets around main sequence binaries remained elusive, so it was unclear whether circumbinary planets—planets that orbit around both stars of a close binary—could be formed. The standard core-accretion model for planet formation struggles to assemble planets close to a binary star due to the large spiral waves launched in the circumbinary protoplanetary disc. In addition, the results from the TATOOINE survey published up to 2010 suggested detecting such planets, if they exist, using radial-velocities is extremely difficult for double-line spectroscopic binaries because of the need to disentangle the two spectra [75]. Soon after, the transit detection of Kepler-16 b [73]—arguably the most surprising discovery from the Kepler mission—marked the first unambiguous detection of a circumbinary planet. The total of 12 transiting circumbinary planets that have since been discovered by the Kepler mission [145] imply an occurrence rate of 10–15% for planets orbiting close binaries stars [146,147]. Konacki et al. [75] suggested that single-lined binaries were likely better suited as targets for radial-velocity surveys for circumbinary planets. The EBLM provided such a sample. Furthermore, thanks to the eclipsing geometry, the true mass of the secondary star is known, ensuring that systems could be selected in which the secondary spectrum is guaranteed to be a negligible fraction of the total flux. This prevents the secondary from interfering with the radial-velocity measurement taken on the primary star's spectrum. A pilot survey was conducted on the southern sample of the EBLM project using the CORALIE spectrograph [44]. Using these results, 100 binary systems were selected and are currently being monitored using the HARPS and SOPHIE spectrographs [148]. Early results from this survey include the detection of Kepler-16 b, using radial-velocity measurements [45], which is the first new planet discovered by the BEBOP project in the system EBLM J0608-59 = TOI-1338 (BEBOP-1 c [8]). This is an external planetary companion to the circumbinary planet TOI-1338/BEBOP-1 b that was identified using the data from the TESS mission [7], but that is yet to be detected using radial-velocity data.

The study of circumbinary planets is becoming an important part of the study of planet formation in general. One goal of BEBOP is to create a sample of circumbinary planets that can be compared with planets around single stars. Such a comparison will help test the robustness and ubiquity of planet formation processes. For example, there is currently some debate about whether super-Earth and sub-Neptune planets are the same exoplanet

population [149,150] and whether these have formed in situ [151] or have instead formed further out and are observed in the present orbits due to disc-driven migration [150]. In situ formation for exoplanets in known circumbinary binary systems is highly unlikely due to a violent truncation and stirring of the inner disc regions [152–154], i.e., these planets almost certainly formed farther out and migrated inwards to their currently observed orbits [155,156]. Circumbinary planet discoveries are currently limited to gas giants and Neptune-like planets [157]. Future surveys with improved photometric accuracy or better radial-velocity measurements will have the sensitivity to find less massive planets that can shed light on these issues. In addition, the properties of exoplanets found near the instability region surrounding the binary stars [152] encodes the properties of the protoplanetary disc that brought them onto the orbits we see them on today [158–160].

The increased spectroscopic monitoring of this subset of EBLM systems is a boon for the EBLM project, increasing the precision of the primary star's parameters (from spectral line analysis [40]) and the parameters of the secondary, but also the possibility to extract the secondary's stars spectral lines and transform some of those systems into double-lined binaries, allowing us to obtain direct mass and radius measurements for both stars [161].

### 5.2. Improved Effective Temperature Measurements

There are two possibilities for improving the accuracy and precision of the $T_{\rm eff,2}$ estimates obtained from our measurements of the secondary eclipse depths in EBLM systems. The first is to directly measure $T_{\rm eff,1}$ from the primary star's bolometric flux and angular diameter derived from its radius and the parallax to the binary measured by the GAIA mission, rather than using an estimate based on spectroscopy. This method requires a direct detection of the M dwarf using high resolution, high signal-to-noise spectroscopy to produce accurate results. This is difficult given the extreme flux ratios in the EBLM systems, but this method was successfully demonstrated for EBLM J0113+31 [36]. The second possibility is to measure the $T_{\rm eff,2}$ directly using measurements of the eclipse depth at multiple infrared wavelengths to obtain the bolometric flux of the M dwarf. This would make it possible to obtain $T_{\rm eff,2}$ with minimal dependence on stellar atmosphere models.

### 5.3. Tidal Evolution

Tides are expected to circularise the orbits of short-period binary stars [162]. The EBLM project observes binary systems with orbital periods in the range from about 2 days up to 10's of days. This spans the canonical circularisation period of $\approx$8 days [64] and also probes circularisation for unequal mass–ratio binaries in which the circularisation is expected to occur on a longer timescale than binaries with mass ratios $\approx$1 due to a weaker tidal influence of the secondary. We used radial-velocity measurements for 118 southern EBLM targets in Paper IV [29] to determine spectroscopic orbits with a median eccentricity precision of 0.0025. This enabled us to find about a dozen binary systems with periods less than 5 days that have small but significant eccentricities ($e > 0.05$). Constraints on the orbital eccentricity and confirmation of the reliability of these small eccentricity values from spectroscopic measurements are now available for many EBLM systems using the data from the TESS mission to measure the phase and width of the secondary eclipse relative to the primary eclipse. Combining all of this information improves our understanding of tidal circularisation timescales, which will be a substantial but worthwhile effort.

Tides are also expected to synchronise the rotation of stars in close binary systems. We expect the M dwarf to be tidally locked to the primary in most of the EBLM systems we have observed (just like the Moon is to the Earth), but in which cases is the primary star also tidally locked? We can use high-resolution spectroscopy to answer this question by measuring the primary star's projected rotation velocity, $v \sin i_\star$. Photometry, e.g., from the TESS mission or the WASP project, can also provide an estimate of the rotation period if the quasi-periodic modulation of the light curve caused by surface inhomogeneities (star spots and plages) can be detected. This method has previously been used to study stellar rotation for large samples of eclipsing binaries, e.g., [66]. The EBLM sample provides

additional information that is complementary to those studies by extending the sample to much more extreme mass ratios so that the tidal influence of the secondary star is weaker. This makes it easier to probe the transition between synchronised and non-synchronised binaries. A complete study in the context of synchronisation timescales is something for a future study, but the analysis of Sethi et al. [163] has derived rotation rates using TESS for over 80 EBLM targets.

One final expectation from tidal evolution is a spin–orbit alignment of the binary. This is measurable using the Rossiter–McLaughlin effect (Section 4.4). We have published results for two EBLM targets and a third star with a brown dwarf companion to date, with all three showing projected spin–orbit angles consistent with zero obliquity [26,31]. Spectroscopy to measure the Rossiter–McLaughlin effect for another two dozen systems has been observed and are currently under analysis. The discovery of spin–orbit misalignment in hot Jupiters was a game changer with respect to understanding their formation and evolution [59,138]. This revolution may await for low-mass eclipsing binaries.

The role of stellar rotation and the consequent magnetic activity on the star has often been discussed in relation to the radius inflation problem [6,22,23,164–168]. One disadvantage of the EBLM project is that there is no direct way to measure the magnetic activity level of the M-dwarf companions. However, it is safe to assume that most of the M dwarfs in EBLM systems rotate synchronously with the orbit, or nearly so, i.e., we can often assume that the M-dwarf's rotation period is equal to the orbital period of the binary. The time scale for the rotation period, $P_{\text{rot}}$, of the low-mass companion to become synchronised with the orbital period, $P_{\text{orb}}$, can be estimated using the following equation [169] (with correction due to A. Barker, priv. comm.)—

$$\tau_\Omega = \frac{2Q'r_g^2}{9\pi} \left( \frac{M_1 + M_2}{M_1} \right)^2 \frac{P_{\text{orb}}^4}{P_{\text{dyn}}^2 P_{\text{rot}}}. \tag{14}$$

Here, $M_1$ and $M_2$ are the mass of the primary star and the low-mass companion, respectively; $r_g \approx 0.2$ is the dimensionless radius of gyration for the low-mass star; $P_{\text{dyn}} = 2\pi / \sqrt{GM_2/R_2^3}$ is the dynamical time scale; and $Q'$ is the tidal quality factor $\sim 10^7 (P_{\text{rot}}/10\text{d})$ for low-mass main-sequence stars. This time scale is much less than 1 Gyr for all EBLM systems with orbital periods of about 10 days or less.

## 6. Discussion

### 6.1. An Updated View of M-Dwarf Properties

The impact of the EBLM project and other efforts to follow up SB1 eclipsing binaries identified using exoplanet transit surveys can be clearly seen by comparing Figure 2 and Figure 7. We have used isochrones from the MIST grid of stellar models at an age of 10 Gyr for these figures [24,25]. In general, the observed radii of fully convective stars ($M_2 \lesssim 0.35 \, M_\odot$) lie within a few percent of the predicted radii. The agreement between the observed and predicted values of $T_{\text{eff},2}$ is also reasonably good if we allow for a possible systematic error $\sim 100$ K in these values due to the reliance on imperfect stellar model atmospheres for these measurements [37]. We have not included measurements with a quoted precision $>5\%$ in this figure. Some of these less-precise measurements do tend to lie well above the main trend in the observed mass–radius, but they are clearly less reliable than the majority of measurements in this compilation. As can be seen in Figure 7, the radius and effective temperature of low-mass stars is also sensitive to metallicity [11]. This is also comparable to the level of disagreement between isochrones generated from different stellar models in this mass range, i.e., the amount by which stars may or may not be inflated depends on which set of stellar models we compare to [11]. EBLMs have the advantage that an estimate of the stars' metallicity is available from the analysis of the primary star's spectrum. It is difficult to estimate the metallicity of very low mass M dwarfs from the analysis of its spectrum, even for field stars, but particularly for rapidly rotating stars in binary systems, so metallicity estimates for M dwarfs in eclipsing binaries are often lacking

and those that exist are likely subject to systematic errors that are difficult to quantify. It is also possible to estimate the age of an EBLM system by comparing the properties of the primary star to stellar models. The stellar density is particularly useful for this age estimated because it is very sensitive to age and can be determined to a high accuracy from the analysis of the light curve [26]. The evolution of main-sequence M dwarfs is negligible within the lifetime of the Universe, but an age constraint is nevertheless useful to rule out the possibility that the M dwarf is a pre-main sequence star that is contracting towards the main sequence.

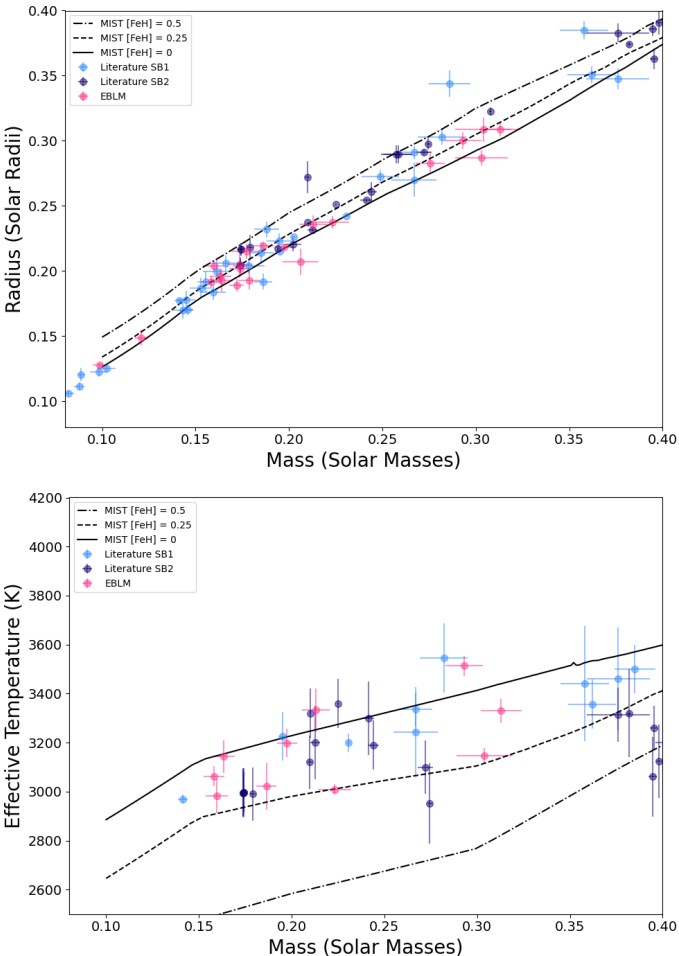

**Figure 7.** Mass–radius and mass–effective temperature relation for very-low-mass stars in SB1 (light blue) and SB2 (dark blue) eclipsing binary systems from the EBLM project and other measurements with quoted precision better than 5% [11,17,26,30–34,36,38,96–99,101,170–210]. The data used to generate this figure are available in the Supplementary Materials, File S1.

### 6.2. On the Accuracy of M-Dwarf Parameters for EBLM Binaries

One criticism of the EBLM project is that the parameters of the secondary stars are dependent on the primary stars' parameters for systems where the analysis is based on the spectroscopic orbit of the primary star only (SB1 systems). Despite the precision on masses and radii of fully convective stars of only a few percent, this prevents the EBLM results from being included in compilations of fundamental data for stars in binary systems, which typically require that the stellar masses are measured directly from the spectroscopic orbits for both stars (SB2 systems) [65,211]. We have used grids of stellar models to estimate the primary star mass in some EBLM papers. These models for solar-type stars are thought to be robust because they are calibrated on the Sun and have been tested against high-quality observations of double-line eclipsing binary stars. Nevertheless, we are moving towards a

more empirical approach to reduce the dependency of our results on stellar models, e.g., by using empirical relations to estimate the primary star mass [122]. In Paper XII [20]), we will show how the stellar radius estimate provided in the GAIA catalogue can be combined with the stellar density obtained from the analysis of the transit to accurately measure the primary star's mass.

A high priority for the EBLM project over the next few years is to transform some of our single-lined eclipsing systems into double-lined systems. The first reason is to demonstrate that the EBLM methods and the precision we claim are both credible by comparing them to double-lined systems, with the idea that our results should be trusted and performed by including the mass–radius relations. The second reason is obviously to obtain absolute dynamical parameters as well because those are interesting in their own right. The first such attempt for EBLM J1031+31 [36] successfully detected the companion spectrum using the data from the near-infrared high-resolution spectrograph SPIROU at CFHT. We will soon publish the detection of the M-dwarf companion to EBLM J0608-59 using an improved methodology applied to the spectra obtained at optical wavelengths with the HARPS and ESPRESSO spectrographs [161]. This improved methodology is adapted from the techniques used to retrieve the spectral signature of exoplanet atmospheres from transmission spectroscopy [212,213]. In the case of EBLM J1031+31, we found that the mass of the secondary measured directly from the stars' spectroscopic orbits ($0.197 \pm 0.003 \, M_\odot$) is fully consistent with the value obtained in Paper II [27] using our standard methodology for SB1 binary systems ($0.186 \pm 0.010 \, M_\odot$) [8,29]. This gives us great confidence that the methods developed in the various EBLM papers produce accurate and precise measurements of very-low-mass stars.

*6.3. Triple Systems*

In Paper IV, we found that 21 out of 118 EBLM systems show evidence for a third star in the systems based on radial-velocity measurements that cannot be modelled as a Keplerian orbit for an isolated binary system, but that are successfully modelled if a linear trend or a second Keplerian orbit are included in the model. The EBLM project was not designed as a study of stellar multiplicity, so this result is difficult to interpret in the context of other studies of the multiplicity properties of solar-type stars [214] because of the strong selection effects that affect this sample. We have avoided these triple-star systems in our observation campaigns to measure the mass, radius, and effective temperatures of the transiting M-dwarf companions because the presence of a "third light" can lead to systematic biases in these measurements [215].

## 7. Conclusions

The EBLM project has been successful in its aims to measure precise masses, radii, and luminosities (effective temperatures) for a sample of very-low-mass stars using observations of single-lined (SB1) spectroscopic binaries containing a solar-type primary star and a transiting M-dwarf companion. The data from this project combined with the results from similar studies gives a substantial body of data for testing models of very-low-mass stars and helps us to fully understand how the mass–radius relation for fully convective stars depends on the composition, orbital period, and rotation rate. The results already show that radius inflation for fully convective stars, if it exists, is a subtle effect amounting to no more than a few percent in the radius. This is comparable to the variation in the stellar radius due to the range of metallicity observed in these systems.

There has also been progress in some of the secondary goals of the EBLM project listed in Section 2. The goal to measure spin–orbit alignment for EBLM systems is the most demanding in terms of scheduling the necessary observations, so results for only two systems have been published so far [26,31], but an analysis is underway for more than 20 systems, so additional results can be expected over the next few years. The intensive radial-velocity monitoring of 118 EBLM targets presented in Paper IV [29] shows that the brown dwarf desert extends into the massive planet regime for short-period systems. The

data in that paper and other papers in the EBLM series are useful for the study of tides using the eccentricity distribution to characterise the efficiency of tidal circularisation (Figure 8). The availability of TESS light curves for many EBLM systems now makes it feasible to also investigate the synchronisation of the primary star's rotation. Work in this area is ongoing, so results can also be expected in the next few years. The goal to seek circumbinary planets has evolved into the BEBOP project [44,148]. This is naturally a long-term project due to the long orbital periods of these planets, but the first successes from this project have been published [8,45] and a substantial observing programme is ongoing, so more discoveries are to be expected in the coming decade.

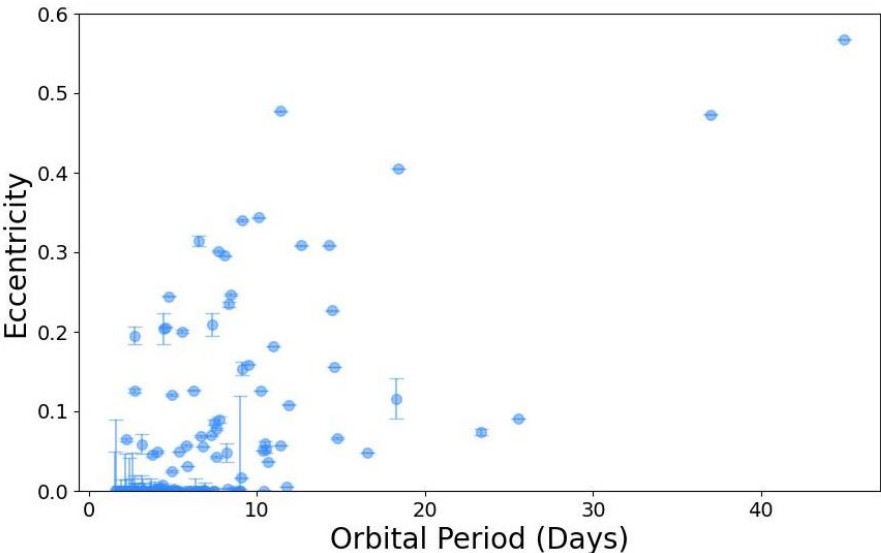

**Figure 8.** Orbital eccentricity as a function of orbital period for EBLM systems [11,20,26,30–34,36].

There is certainly scope to improve the quality of the data available for the very-low-mass stars shown in Figure 7 using the methods we have developed during the EBLM project. For example, many of the stars shown lack metallicity measurements. It would also be helpful to use a consistent and accurate method to estimate the primary star parameters for the SB1 systems in Figure 7, e.g., using the radii derived from the GAIA mission. We can also look forward to a sample of EBLM systems for which the primary star parameters will be derived using asteroseismology from the PLATO mission [216].

**Supplementary Materials:** The following supporting information can be downloaded at: https://www.mdpi.com/article/10.3390/universe9120498/s1, File S1: Mass, radius, and effective temperature measurements for very-low-mass stars in eclipsing binary systems.

**Author Contributions:** Conceptualisation, P.F.L.M.; writing—original draft preparation, P.F.L.M., A.H.M.J.T. and D.V.M.; writing—review and editing, P.F.L.M., A.H.M.J.T. and D.V.M.; visualisation, P.F.L.M. and D.V.M. All authors have read and agreed to the published version of the manuscript.

**Funding:** P.F.L.M. was funded by the Science and Technology Facilities Council (STFC), grant numbers ST/S001301/1 and ST/R000638/1. AHMJT is supported from the European Research Council (ERC) under the European Union's Horizon 2020 research and innovation programme (grant agreement n° 803193/BEBOP). The EBLM project has received generous funding and observing time allocations from NASA through TESS Guest Investigator Programs G05024, G04157, and G022253 (PI Martin).

**Data Availability Statement:** The compilation of mass, radius, and effective temperature measurements shown in Figure 7 are available in the supplementary data that accompany this article.

**Conflicts of Interest:** The authors declare no conflict of interest.

## Abbreviations

The following abbreviations are used in this manuscript:

| | |
|---|---|
| EB | Eclipsing binary (star) |
| EBLM | Eclipsing binary–low mass |
| CLV | Centre-to-limb variation |
| RM | Rossiter–McLaughlin |
| RV | Radial velocity |
| SB1 | Single-lined spectroscopic binary star |
| SB2 | Double-lined spectroscopic binary star |

## Notes

1   It later transpired those small eccentricities were spurious, artefacts of the fitting algorithms which were later adapted to avoid the problem [56].

2   https://wasp-planets.net (accessed on 19 November 2023).

3   This system was assigned a WASP number rather than an EBLM identifier because it has a "sub-stellar" companion.

4   Also referred to as $\beta$ where $\beta = -\lambda$.

5   https://github.com/pmaxted/ellc (accessed on 19 November 2023).

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
