# Peer review of "The EBLM Project—From False Positives to Benchmark Stars and Circumbinary Exoplanets"

_universe, doi:10.3390/universe9120498_

Round 1
Reviewer 1 Report
Comments and Suggestions for Authors
I am happy you pointed out (first paragraph of section 6.2) the model dependency for masses in SB1 systems. To get to completely model-independent absolute parameters, we need SB2 systems, and hopefully, we are slowly getting there as observing techniques evolve. Nice review paper!
Author Response
We thank the reviewer for the positive comments on our manuscript.
No changes necessary.
Reviewer 2 Report
Comments and Suggestions for Authors
The author reviewed the EBLM project, with its history, motivation, method, and some results. The paper is well written and the methods developed to detect those small stars are impressive, the scientific results are good. But there are major and minor problems that need to be corrected.
Major problems:
In Section 2, Goals of the EBLM project. The main results are discussed in Section 6. But as the other goals listed on pages 5-6 are more like a prespect of the EBLM project. Although we know both the spectra and lightcurve follow-ups are very time-consuming. But as a review paper, after the project ran for 10 years, the audience would expect to see how many more times are needed to achieve those goals from your experience.
Section 4.4 and 4.5 are more related to Section 5.3 tidal evolution than Section 4 Methodology, I would suggest to put those together.
Minor problems,
1. page 3, fig 1, since there is no EBLM object in this figure, it is prefered to remove the EBLM symbol in the figure.
2 page 10, line 339, period shorter than 2 days ... so were "not" flagged as EBLM systems.. I think the"not' here should be a typo?
3.page 11, equation 9, as the author metioned e and w in the previous sentence, indicating not a circular orbit. the mass function should include ellipticity e.
4.page 11 line 361, It not obvious how a 10% mass error will lead to <1% error in rho, it hard to understand how the error propagated, please explain.
5.page 13 line 459, "also found the the" here is a typo.
6 page 16 line 588 vsinI1, it is better to use lowercase vsini1
Author Response
We thank the reviewer for their careful reading of the manuscript and
thoughtful comments that have helped to improve the paper.
> In Section 2, ... how many more times are needed to achieve those goals
> from your experience.
Added a paragraph to Section 7 with a summary of expectations for the
secondary goals of the project as far as can be foreseen at present.
> Section 4.4 and 4.5 are more related to Section 5.3 tidal evolution than
> Section 4 Methodology, I would suggest to put those together.
We agree that content in section 4.5 belongs in section 5.3 so we have
combined these two sections (new section 5.3). Section 4.4 is a description of
the methods we have used to measure spin-orbit alignment so we prefer to
leave this in section 5 together with the descriptions of the other methods we
have used.
> 1. page 3, fig 1, since there is no EBLM object in this figure, it is
> prefered to remove the EBLM symbol in the figure.
Done.
> 2 page 10, line 339, period shorter than 2 days ... so were "not" flagged
> as EBLM systems.. I think the"not' here should be a typo?
No. Rephrased this sentence to clarify this point - "and so were typically
flagged as eclipsing binaries or some other variable type, not as EBLM
systems."
> 3.page 11, equation 9, as the author metioned e and w in the previous
> sentence, indicating not a circular orbit. the mass function should include
> ellipticity e.
Done.
> 4.page 11 line 361, It not obvious how a 10% mass error will lead to <1%
> error in rho, it hard to understand how the error propagated, please
> explain.
Added "because the mass ratio is weakly dependent on the assumed primary star
mass if $M_1\gg M_2$ and the mass ratio only enters the equation for the
stellar density via the term $(1+q)$. "
> 5.page 13 line 459, "also found the the" here is a typo.
Fixed.
> 6 page 16 line 588 vsinI1, it is better to use lowercase vsini1
Replaced with v.sin i_star for consistency with previous notation in the
manuscript for this quantity.
Reviewer 3 Report
Comments and Suggestions for Authors
The paper is well written and may be recommended in a present form. A review of the previous results is enogh complete. There are minor "cosmetic" remarks, like the criterion to choose the interval width in Fig. 1. It may be chosen e.g. by computing such means for differnet widths and the best r.m.s. accuracy estimate may be such a criterion. Also one may use some phenomenological patterns/shapes, like in the "New Algol Variable" algorithm (e.g. applications of this method in 2016JPhSt..20.4902T . As both components are of stellar nature, these systems are algols. And phenomenological approximation may give better acuracy for the period determination, prior to make a physical model. However, this is just for discussion, not absolutely necessary.
Other small remark like GAIA shoul be in capital letters, as the abbreviation in the bold font, not Gaia in italics.
Author Response
We thank the reviewer for their comments on the manuscript.
We decided not to add a discussion of the best algorithm to use for the
period determination from the data shown in Fig. 1 because the WASP data are
now of mostly historical interest only given the availability of TESS light
curves and high-precision radial velocity measurements for the majority of
EBLM systems.
> Other small remark like GAIA shoul be in capital letters, as the
> abbreviation in the bold font, not Gaia in italics.
Done.
Reviewer 4 Report
Comments and Suggestions for Authors
The review paper by Maxted et al. “The EBLM Project – From False Positives to Benchmark Stars and Circumbinary Exoplanets” summarizes the long-term EBLM project. I think that the topic and the way it is presented nicely fit the aims of the Special Issue "The Royal Road: Eclipsing Binaries and Transiting Exoplanets". I found the paper complete and well written, including the “historical” perspective describing the origin of the project and how it evolved and the emerging results. The review also includes basic relationships for close binary systems and descriptions, making it an appropriate reference for not fully expert readers.
I have a few specific comments that I ask or I suggest to be addressed by the authors.
- Ensemble figures are presented only for Mass-radius and Mass-Teff relationships. I think also the period-eccentricity (or similar) should be presented. [In case the paper becomes too long Fig. 2 and Fig 7 could be merged presenting the few old data points (known at the time the project started) with different symbols/colors]
- Figure 7 shows the theoretical models for three different metallicities: however, it is not clear to which metallicity the observational data should be compared. I understand the usefulness of showing with different colors the different types (SB1 vs SB2) or sources (EBLM vs others) of the binaries in the sample. One suggestion to combine both kind of information in the same figure is to use some color code for metallicity and different symbols for the types/sources of the targets.
- Close binaries are very often found to have additional companions in wider orbits (see. e.g. Tokovinin et al. 2006), with a dependence of the third companion fraction with the central binary orbital period. What is the fraction of the investigated systems with known additional companions (e.g. from Gaia, dedicated imaging searches, RV monitoring, etc.) ? How much is this census complete? Can unrecognized third light effects play a significant role in the astrophysical results discussed?
- [suggestion] In the long-term perspective (e.g. PLATO) it would be useful to understand how much the radial velocity follow-up is still needed for low-mass binaries in the present (and future) time considering the spectroscopic monitoring performed by Gaia. In this context, I suggest to check the fraction of binaries for which the RV orbit is correctly retrieved from Gaia RVs (NSS catalog).
- Supplementary material: Explanations of the columns. Bytes 47- 51. Stellar radius → Stellar mass
Author Response
----------
We thank the reviewer for their careful reading of the manuscript and
thoughtful comments that have helped to improve the paper.
> I think also the period-eccentricity (or similar) should be presented.
Done
- Figure 7 .. use some color code for metallicity
We have experimented with this option but found that the resulting figure is
very difficult to interpret, partly because of the complexity, but also
because the metallicity measurments from other studies are often missing and
have a wide range of uncertainities, both random and systematic.
- Close binaries ..
We have added a subsection on the point but have kept it short because little
work has been done on stellar multiplicity within the EBLM project and we do
not feel it is appropriate to add new work on this important subject in a
review paper. .
- Supplementary material: Explanations of the columns. Bytes 47- 51. Stellar
radius → Stellar mass
Fixed.